# Berry Fruits and Their Improving Potential on Skeletal Muscle Health and Performance: A Systematic Review of the Evidence in Animal and in Human Studies

**DOI:** 10.3390/foods13142210

**Published:** 2024-07-13

**Authors:** Alessia Moroni, Roberta Zupo, Fabio Castellana, Federica Amirante, Marco Zese, Mariangela Rondanelli, Patrizia Riso, Simone Perna

**Affiliations:** 1Endocrinology and Nutrition Unit, Azienda di Servizi Alla Persona “Istituto Santa Margherita”, University of Pavia, 27100 Pavia, Italy; alessia.moroni02@universitadipavia.it (A.M.); marco.zese01@universitadipavia.it (M.Z.); 2Department of Interdisciplinary Medicine (DIM), University of Bari Aldo Moro, Piazza Giulio Cesare 11, 70100 Bari, Italy; fabio.castellana@uniba.it (F.C.); f.amirante@studenti.uniba.it (F.A.); 3Department of Public Health, Experimental and Forensic Medicine, University of Pavia, 27100 Pavia, Italy; mariangela.rondanelli@unipv.it; 4Department of Food, Environmental and Nutritional Sciences, Division of Human Nutrition, University of Milan, 20133 Milan, Italy; patrizia.riso@unimi.it (P.R.); simone.perna@unimi.it (S.P.)

**Keywords:** berries, skeletal muscle, performance, blackcurrant, blueberry, raspberry, aronia, elderberry

## Abstract

The well-established anti-inflammatory and antioxidant properties of red fruits leave room for a biological pathway of improved muscle health promoted by berries in the diet. Our objective was to systematically review the number of trials conducted on human and animal species around the relationship between a berry diet and muscle health outcomes. Two independent examiners conducted a search for studies that utilized keywords associated with muscle health outcomes and a berry-based diet in both human and animal trials, in accordance with the PRISMA statement guidelines. The literature was searched through six electronic databases until December 2023. Screening of 152 retrieved articles resulted in a final selection of 16 reports investigating the effect of exposure to a berry-based diet and skeletal muscle health outcomes. The study protocol was registered on PROSPERO (CRD42023479682). Among the selected studies, nine involved humans and seven animal models (rats and mice). Overall, most of the studies reported positive effects on performance or muscle health. Specifically, five studies investigated the possible effects of blackcurrant on active human subjects or athletes; three studies focused on blueberry and presented results on running performance (human sample) and muscle health (rat models). The rest of the studies involved raspberries (two studies, rat models), aronia (one study, rat models), elderberry (one study, rat models), and a mixed compound (one study, rat models). In conclusion, there is some early evidence that a berry-rich diet may increase performance or muscle health, but more research is needed to fully understand the underlying biological trajectories, and thus, no firm conclusions can yet be drawn.

## 1. Introduction

Consuming a diet high in fruits has been consistently linked to a lower chance of developing chronic illnesses because fruits include a number of compounds that can protect health in addition to vitamins and minerals, including antioxidants and anti-inflammatory compounds [1]. Wild plant species are intriguing to the food business because of their capacity to replace synthetic chemicals and nutraceuticals, but the nutritional, economic, and sociocultural worth of these natural resources has not been properly researched and utilized. Nevertheless, the relationship between diet and health holds much interest nowadays among consumers, leading researchers to seek more information on diets rich in nutraceuticals, including fruits and vegetables [2,3,4,5]. Their bioactive constituents have beneficial effects on human health in terms of prevention from chronic diseases and improvement of life quality.

Berries may be one of the components of healthy diets, as they contain a wide variety of phenolic compounds [6,7]; blackberry (*Rubus* sp.), blueberry (*Vaccinium myrtillus*), black currant (*Ribes rugrum*), chokeberry (*Aronia melanocarpa*), cranberry (*V. macrocarpon*), raspberry (*R. ideaus*), black raspberry (*R. occidentalis*), and strawberry (*Fragaria ananassa*) are usually consumed in fresh and processed forms in the human diet; interestingly, berries are exceptionally rich sources of phenolic antioxidants [8,9]. Phenolic compounds are the main group of phytochemicals in berries, including flavonoids (anthocyanins, flavonols, flavones, flavanols, flavanones, and isoflavonoids), stilbenes, tannins, and phenolic acids [10].

Of the several beneficial health-promoting effects reported so far in the literature, a certain amount of literature leaves room to hypothesize that fruit-derived polyphenols—and especially dark-colored fruits—may work well in improving physical exercise performance [11], considering that increased production of reactive oxygen species (ROS) has been linked to the onset of fatigue and that inflammation and oxidative damage in muscles aid in the recovery process following vigorous activity. In this context, a moderate amount of both human and animal literature has so far investigated the effects of a berry diet and/or berry extract(s) supplementation on outcomes of physical recovery [12,13] and performance as well as other domain(s) of muscle function [12] and health. Yet, an overview of the evidence on the effect of a berry diet (or supplementation with extracts derived from berry fruits) on skeletal muscle health outcomes is still lacking.

The aim of this systematic review was to examine the amount of literature around the relationship between a “berry” diet, which includes the consumption of “red fruits” such as strawberries, currants, gooseberries, blackberries, raspberries, and cranberries, and any domain of skeletal muscle health and performance (i.e., training speed, physical performance, muscle circulation, fatigue, recovery, etc.) in both human and animal species.

## 2. Materials and Methods

### 2.1. Search Strategy, Study Selection, and Data Extraction

A computerized literature search of MEDLINE and the Cochrane database did not identify any previous systematic reviews on exposure to a berry diet and/or berry extract and domain(s) of skeletal muscle health in both animal and human species. This systematic review adhered to the PRISMA 27-item checklist and the Preferred Reporting Items for Systematic Reviews and Meta-Analyses (PRISMA) standards [14]. The flow of the literature screening process is reported in Figure 1. An a priori protocol for the search strategy and inclusion criteria was established and recorded, with no particular changes to the information provided at registration on PROSPERO, a prospective international registry of systematic reviews (CRD42023479682). We performed separate searches in the U.S. National Library of Medicine (PubMed), Medical Literature Analysis and Retrieval System Online (MEDLINE), EMBASE, Scopus, Ovid, and Google Scholar to find human and animal trials evaluating the effect of a berry diet (or berry extracts) on muscle health domain(s) (i.e., training speed, physical performance, muscle circulation, fatigue, recovery, etc.). Therefore, the main goal was to assess whether the abundance of berries (i.e., wolfberry, raspberry, blackcurrant, black chokeberry, goji berry, elderberry, and blueberry) could benefit general muscle health. We also considered grey literature using the huge archive of preprints https://arxiv.org/ (accessed on 31 December 2023) in the study selection phase and the database http://www.opengrey.eu/ (accessed on 31 December 2023) to access abstracts of notable conferences and other unreviewed material.

In order to include research into the analysis, the following criteria were used: (1) randomized controlled clinical trial in humans or animals with a crossover or parallel design; (2) main findings on the effect of each berry supplementation on skeletal muscle health domain(s); (3) conference abstracts and nonclinical trial studies were not included.

The search approach applied in PubMed and MEDINE and modified to the other four electronic sources contained keywords such as “berr*”, “skeletal muscle”, and each type of berry combined through the use of Boolean indicators such as AND and OR (Table 1). The search strategy used the Boolean indicator NOT to exclude opinion papers, letters, reviews, and meta-analyses. The literature search had no time restrictions, and papers were retrieved until 31 December 2023. Two researchers (A.M. and M.Z.) examined the papers individually and in duplicate, evaluated the complete texts, and chose the articles for inclusion into the study after reviewing the titles and abstracts of the obtained publications. Inter-rater reliability (IRR) was applied to project inter-coder agreement and subsequently the κ statistic to assess accuracy and precision. In all data extraction processes, a coefficient k of at least 0.9 was attained, per PRISMA principles and the quality evaluation procedures [15].

### 2.2. Quality Assessment

Using the revised Cochrane risk-of-bias tool for randomized trials (RoB-2), which is comprised of five main domains, including bias arising from the randomization process, bias due to deviations from the intended interventions, bias due to missing outcome data, bias in the outcome measurements, and bias related to the selection of reported results, two reviewers (R.Z. and F.C.) independently evaluated the methodological quality of the eligible RCTs. Final opinions and general risk of bias were characterized as “Low” or “High” risk of bias or stated as “Some Concerns” [16]. A third reviewer (R.Z.) checked this assessment.

## 3. Results

The first systematic literature search yielded 152 entries. After excluding duplicates, 75 were classified as potentially relevant and selected for title and abstract analysis. Then, 37 were excluded because they did not meet the characteristics of the approach or the objective of the review. After reviewing the full text of the remaining 38 papers, only 16 met the inclusion criteria and were included in the final qualitative analysis [13,17,18,19,20,21,22,23,24,25,26,27,28,29,30,31].

The Preferred Reporting Items for Systematic Reviews and Meta-analyses (PRISMA) flow chart illustrating the number of studies at each stage of the review is shown in Figure 1.

The final study base included 16 articles reporting on the effect of a berry diet (or berry extracts supplementation) on muscle health domain(s) (i.e., training speed, physical performance, muscle circulation, fatigue, recovery, etc.).

Details of the study design, population, country, author(s), year of publication, exposure, intervention timing, outcome(s), and main findings are provided in Table 2.

The risk of bias was assessed and found to be low in 4 of the 16 studies [22,27,30,31] and moderate in 8 studies [13,17,18,19,20,21,23,24], while the other 4 raised more concerns, mostly in domains 2, 3, and 4 [25,26,28,29] of the RoB-2 tool (Figure 2)

The included studies were equally distributed between humans (9 out of 16) and animals (7 out of 16). The geographic distribution of studies favored Europe (7/16, 44%), followed by Asia (5/16, 31%), the USA (3/16, 19%), and a minority in New Zealand (1/16, 6%). The timing of intervention was found to be inconsistent across the selected studies, varying from a minimum of a 7 days to a maximum of 8 weeks.

Muscle health outcomes were shown to be several among those considered in the selective studies, and some of the domains included evaluations of muscle functioning as follows: agility testing, seated medicine ball throwing, hand grip strength testing, assessment of power output, muscle oxygen saturation, fatigue, muscle soreness, maximum voluntary contraction (MVC), submaximal forearm muscle contractions, force of contraction during recovery, incremental running test, timed performance over 5 km, contractility and sensitivity of vascular smooth muscle function and muscle metabolism, and motor failure and muscle incoordination, among others. Findings on muscle health and function in relation to exposure to different berry fruit classes are detailed as follows.

### 3.1. Blackcurrant

Studies concerning supplementation with blackcurrant-based compounds were globally 10, all conducted in human samples. Seven studies were conducted in the United Kingdom (U.K.), two in New Zealand, and one in Japan, from 2005 to 2022. Samples ranged from 9 to 27 participants (aged from 21 to 39 years old), and the majority of the studies regarded trained subjects, including professional athletes of different disciplines such as rugby [17], rock-climbing, recreational running [22,26], and non-resistance-training sports [20]. Three studies involved healthy or active participants with no reference to a specific training activity [21,25,29]. Most of the studies involved male participants, while a few studies involved either males or females [20,22]. Targeted outcomes concerned mainly performance variables but also soreness, fatigue, and blood circulation. The results showed increases in performance speed and agility in rugby participants without changings in muscle strength. The study on climbing concerned only men, demonstrating an increase in time to half recovery (TTHR) [23]. No effects were reported for performance variables. Studies focused on running involved females only [26] and both males and females [22], showing a possible improvement on peak running speed. Urine IL-6 levels rose in the NZBC group 48 h after the half-marathon (*p* < 0.01) but remained stable in the placebo group (*p* > 0.05). Perceived muscle soreness and fatigue increased immediately after the half-marathon (*p* < 0.01) and reverted to pre-marathon levels within 48 h, with no difference between groups (*p* > 0.05). Regarding muscle health, Hunt et al. [20] showed how consumption of NZBC extract prior to and following a bout of eccentric exercise attenuates muscle damage and improves functional recovery. Cook et al. [25] investigated muscle oxygen saturation, muscle activity, and femoral artery diameter of the quadriceps, showing how femoral artery diameter increased with intake of NZBC extract during a submaximal (i.e., 30% iMVC) 120 s sustained isometric contraction of the quadriceps muscles. The enlarged diameter of the femoral artery was accompanied by alterations in cardiovascular responses with a decrease in systolic and diastolic blood pressure, mean arterial blood pressure, and total peripheral resistance, with a concomitant increase in cardiac output and stroke volume. The only study regarding cycling reported no significant results [22,26]. Vitamin C and polyphenols, particularly anthocyanins (476 mg/100 g), are prevalent bioactives in blackcurrants. From a functional point of view, consumption of anthocyanin-rich foods is proven to decrease loss of muscle strength after exercise. At the same time, antioxidant vitamin supplementation has been used to prevent oxidative stress and improve performance and muscle mass.

### 3.2. Blueberry

Studies concerning supplementation with blueberries were three. One of them was conducted on human samples and two on animal samples. Regarding the study concerning human protocol, Brandenburg and colleagues [24] studied the effect of four days of supplementation with blueberry on 14 runners during an 8 km race. The supplementation did not alter the race performance but had a positive effect on blood lactate levels measured after the race: (5.4 ± 2.0 mmol/L) in comparison with the placebo group (6.6 ± 2.5 mmol/L; *p* = 0.038). Two animal case–control studies were conducted on rats. Both studies investigated the role of blueberry on the vascular smooth muscle contractile machinery. Norton and colleagues [28] found that a wild blueberry diet affects the vascular tone by suppressing the alfa 1-adrenergic receptor agonist-mediated contraction. Rats were fed with a blueberry powder that was incorporated at 8% in the diets. After 13 weeks, the endothelium maximum force of contraction (Fmax) in blueberry-fed animals was lower (0.9266 ± 0.0463 g) if compared with the control diet (1.109 ± 0.0463 g) (*p* < 0.05). In the second study, the authors questioned the timing needed to make a blueberry-based diet effective on smooth muscles. After trying various protocols, Del Bo’ and colleagues [27] concluded that blueberries incorporated into the diet at 8% w/w at least for 7 weeks can improve endothelial function ameliorating vascular tone. The bioactive potential of blueberries is certainly attributable to their high vitamin C content: 100 g of blueberries provide, on average, 10 mg of ascorbic acid, which is 1/3 of the recommended daily intake. In addition, the total polyphenol content in blueberries varies from 48 to 304 mg/100 g fresh fruit weight (up to 0.3%) and is strictly dependent on cultivar, growing conditions, and maturity. The polyphenols present in blueberries include flavonoids, procyanidins (in monomeric and oligomeric forms), flavonols (e.g., kaempferol, quercetin, and myricetin), phenolic acids (mainly hydroxycinnamic acids), and stilbene derivatives. During the ripening of blueberries, a shift in the pool of total polyphenolic compounds toward anthocyanin synthesis was observed, in line with the decline of the other individual phenolic components, suggesting their important role in blueberry bioactivity. It has been reported that anthocyanin content varies from 25 to 495 mg/100 g of blueberries and depends on fruit size, stage of ripeness, as well as climatic, pre-harvest environmental and storage conditions.

### 3.3. Mixed Berries/Other

#### 3.3.1. Raspberry

Studies concerning raspberry compounds were only two, with different study designs and outcomes. Both studies were conducted in animals (male rats). Choi et al. [19] led RCT research targeting acid-induced hyperalgesia and demonstrated how raspberry (immature *R. occidentalis*) demonstrated antinociceptive properties against acid-induced chronic muscular discomfort. Its activity may be mediated by the α2-adrenergic, nicotinic cholinergic, and opioid receptors. iROE has a higher antinociceptive tendency than mature ROE. Shukitt-Hale et al. [13] focused on baseline motor performance in a case–control study. The study revealed that poor performers fed with 1% or 2% raspberry had higher post-test composite scores (*p* < 0.05), whereas strong performers fed with 2% raspberry had lower post-test composite scores (*p* < 0.05) compared to controls. In plank walking, 1% and 2% raspberry sustained good performance and improved poor performance (*p* < 0.05), respectively. Additionally, 2% raspberry raised poor performers’ grip strength post-test. Additionally, rats with lower post-diet composite scores had higher blood IL-1β levels (r = −0.347, *p* < 0.05). Anthocyanins rank amongst the most essential active components of raspberries and are water-soluble natural pigments with strong resistance to oxidation, to which the greatest antioxidant potential of raspberries is attributed. Black raspberries have been found to possess a higher anthocyanin content (up to 400 mg/100 g) than blackberries (<150 mg/100 g), followed by red raspberries (20–60 mg/100 g), orange raspberries (0.3–8.7 mg/100 g), and finally yellow raspberries (0–3.4 mg/100 g).

#### 3.3.2. Aronia

Chokeberry, also known as aronia (AR), was studied for its myogenic differentiation and muscle metabolic functions using young mice as a model [30]. To investigate the effect of aronia, mice received a diet containing a daily dose of 3.3 mg/kg AR powder for 8 weeks. Biopsy and muscle functional analysis were performed to compare the AR diet and control diet. Regarding the muscle mass, the authors found that the weights of the soleus and extensor digitorum longus muscles, two of the hindlimb muscle groups, were increased approximately 20.0 and 15.3% in mice fed with aronia compared with the control mice. Regarding muscle function and muscle strength, mice fed with aronia showed a 14.5% increase in the grip strength test compared with the control group. These data suggest that a diet enriched with aronia improves muscle mass and muscle function in mice models. Black aronia is rich in phenolic substances, especially proanthocyanidins (total content can reach 4790 mg/100 g FW), anthocyanidins, and other flavonoids as well as phenolic acids. These bioactive compounds provide the potential health benefits of black aronia.

#### 3.3.3. Elderberry

Elderberry (EB) is known to show antioxidative and anti-inflammatory action and to show neuroprotective properties. One study [31] was designed to understand the protective role of EB in rats treated with a neurotoxin agent, 3-nitro propionic acid (3-NP). Rats were fed for 8 weeks with a diet containing the neurotoxin agent, with (3-NP + EB) or without (3-NP) 2% of elderberry; subsequently, outcomes such as motor coordination, locomotion, and neuromuscular activity were analyzed and compared with the control group. Rotarod performance test and electromyography indicated that EB treatment recovered muscle incoordination and motor failure in rats treated with the neurotoxin agent. The elderberry flower was found to be the richest source of total polyphenols (10.04%) and phenolic acids (2.90%); the leaf the richest in flavonoids (1.48%), chlorophylls, and carotenoids (931.7 and 133.77 mg·100 g^−1^ dry matter, DM, respectively); and the fruit in anthocyanins (0.29 g·100 g^−1^ DM).

#### 3.3.4. Wolfberry/Mixed Compounds

Fatigue and exercise performance are strictly correlated. To explore the potential antifatigue role of plant extracts and their components, one study prepared a formula composed of wolfberry, figs, white lentils, raspberries, and maca (WFWRM) [18]. Outcomes such as exercise capacity and antioxidant markers were investigated in mice exhaustively exercised through specific tests. Mice were fed for 30 days with a WFWRM powder (1.00 g/kg of body weight). At the end of the feeding period, mice showed increased exercise ability. WFWRM significantly increased rotarod time by 147.07% compared to the model group (*p* < 0.001). Furthermore, cytosolic markers were used to assess muscle injury and hence fatigue. Compared to the control, the levels of LA, CK, and LDH decreased, respectively, by 34.0%, 43.5%, and 33.5% in WFWRM (*p* < 0.0001). These data indicate that WFWRM formula has antifatigue activity.

## 4. Discussion

The purpose of this study was to systematically review the literature and provide an overview of the possible effects of a berry-enriched diet (i.e., consumption of red fruits such as strawberries, currants, gooseberries, blackberries, raspberries, and blueberries) on skeletal muscle health and performance across trials conducted in both human and animal species. We included 16 papers, 9 of which addressed human research and 8 animal models. As for berry compounds, most of the studies involved protocols using blackcurrant supplements/compounds, mainly involving active subjects/samples or sports professionals. There were only three studies on blueberries, and again, the only research involving humans involved athletes (runners [24]), while the other studies reported positive results for muscle health obtained in animal samples [27]. The remaining included studies focused on raspberries [13,19], which reported positive results in terms of performance and health; aronia [30], with findings of increased grip strength and muscle mass; elderberry [31]; and, finally, a mixed composite supplement including wolfberry, fig, white lentil, raspberry, and maca (WFWRM) [18] that showed positive effects on exercise capacity and fatigue. Despite the heterogeneity of protocols and samples (even regarding the same berry or even the same compound), all studies reported positive results on muscle health or performance, suggesting how targeted consumption of these fruits can lead to positive health outcomes.

When analyzing the findings on performance and thus mainly on the effects of blackcurrant, the authors showed improved speed and agility performance [17], possible improvements in peak running speed (runners) and exercise capacity, and overall performance in animal models (respectively: mice, gooseberry/mixed compound; rats, raspberry). Regarding grip strength, aronia yielded positive results on mice models [30], while in humans, the only study that included such parameters yielded no changes after the consumption of the blackcurrant supplement [17]. Studies that investigated skeletal muscle health-related effects reported an increase in time to half recovery ((TTHR) in climbers, blackcurrant), attenuation in muscle damage and improvement on functional recovery (humans, blackcurrant), increase in artery diameter of quadriceps during a submaximal contraction (humans, blackcurrant), positive effects on blood lactate after a 8 km running race (humans, blueberry [24]), improvements in endothelial functions and thus in vascular tone (rats, blueberry), antinociceptive activity against acid-induced chronic muscle pain (rats, raspberries), increase in muscle mass (mice, aronia [30]), recovery in muscle incoordination and motor failure when exposed to a neurotoxin (rats, elderberry), and antifatigue activity (mice, mixed compound [18]).

Although most studies reported positive and encouraging results, it must be recognized that specific conclusions or even guidelines cannot be drawn at present, either on single berries/single compounds or on mixed compounds, as the literature is still growing. Indeed, further studies on human samples and on specific protocols and compounds are needed to obtain better information on the different mechanisms of action of each berry and thus address the specific result in a context-driven way. To date, some underlying mechanisms have only been postulated and argue that antioxidant supplementation may improve endurance training, antioxidant capacity in the immediate post-exercise period, mitochondrial biogenesis, cellular defense mechanisms, and insulin sensitivity. Antioxidants may also interfere with muscle growth by inhibiting muscle cells’ ability to repair their cell membranes following intense loading.

To our knowledge, this is the first systematic review that has investigated the relationship between a berry diet and its possible effects on health and muscle performance, which we believe is the main strength of this study. Limitations include the lack of comparison between studies on the same berry and similar samples/models, the small number of studies analyzed, and their heterogeneity in quality and the inconsistency of results. To our knowledge, this is the first systematic review that has investigated the relationship between a berry diet and its possible effects on health and muscle performance, and we believe this is the main strength of this study.

In summary, considering the positive results on performance or muscle health, the studies suggest that a berry-enriched diet could lead to improvements, but we cannot yet draw specific hypotheses, as the research needs further studies on berries/compounds and specific samples.

## 5. Conclusions

There is early literature showing that a berry-rich diet can increase performance or muscle health, but more research is needed to fully understand the underlying biological trajectories, and thus, solid conclusions cannot yet be drawn.

## Figures and Tables

**Figure 1 foods-13-02210-f001:**
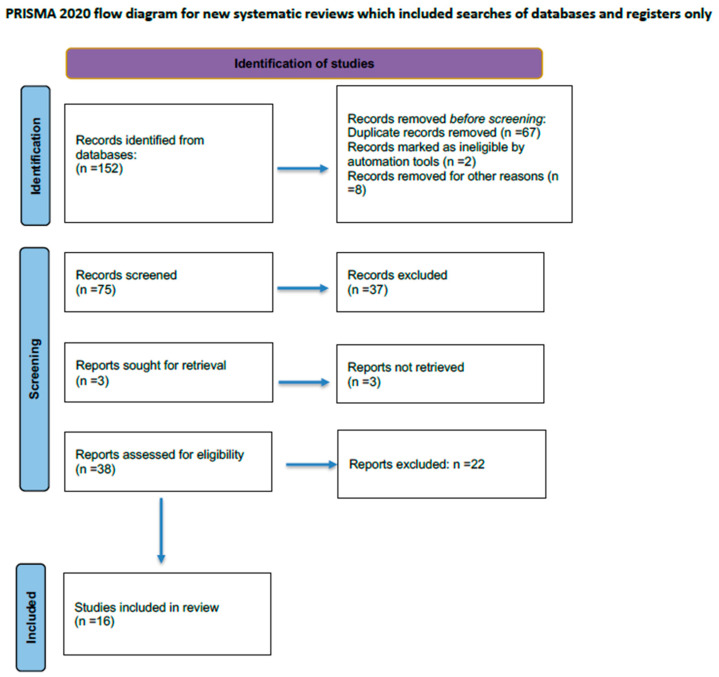
Prisma flowchart of the screening process.

**Figure 2 foods-13-02210-f002:**
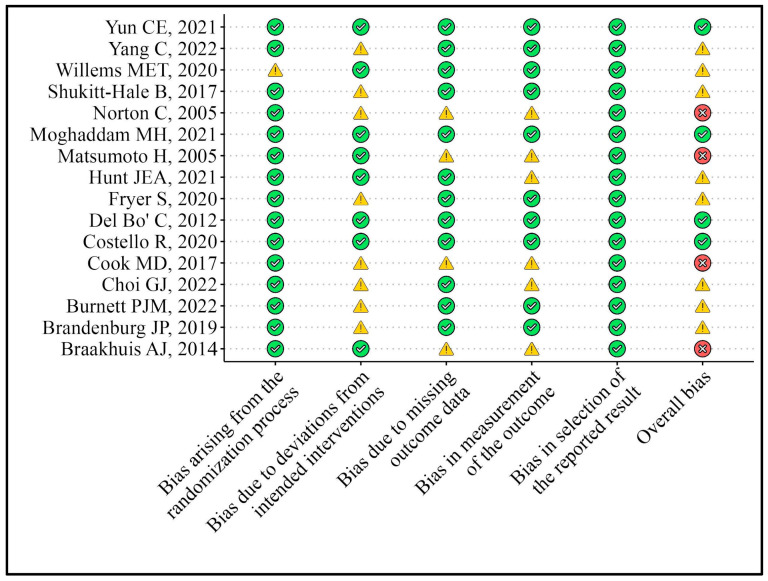
Quality assessment of the studies [13,17,18,19,20,21,22,23,24,25,26,27,28,29,30,31].

**Table 1 foods-13-02210-t001:** Search strategy used in the US National Library of Medicine (PubMed) and Medical Literature Analysis and Retrieval System Online (MEDLINE) and adapted to the other sources, according to selected descriptors.

	*Strategy*	*Descriptors Used*
#1	Population	(Human[tiab]) OR (Animal*[tiab])
#2	Intervention/Exposure	(wolfberry[tiab]) OR (raspberry[tiab]) OR (blackcurrant*[tiab]) OR (chokeberry*[tiab]) OR (goji*[tiab]) OR (elderberry*[tiab]) OR (blueberry*[tiab])
#3	Comparator	(supplement*[tiab]) OR (exposure[tiab]) OR (level*[tiab])
#4	Outcomes	(skeletal muscle[tiab]) OR (muscle health[tiab]) OR (muscle[tiab]) OR (speed[tiab]) OR (physical performance[tiab]) OR (muscle circulation[tiab]) OR (fatigue[tiab]) OR (recovery[tiab])
#5	*Exclusion keywords*	(Review[tiab]) OR (systematic review[tiab]) OR (narrative review[tiab]) OR (meta-analysis[tiab]) OR (editorial[tiab]) OR (letter[tiab]) OR (commentary[tiab]) OR (perspective[tiab]) OR (book[tiab])
#6	*Search strategy*	#1 AND #2 AND #3 AND #4 NOT #5
	Filters: Sort by: Most Recent. Date: 11 December 2023. Time restriction: none.	

**Table 2 foods-13-02210-t002:** Descriptive of human and animal studies selected for inclusion criteria. *N* = 16.

Author, Year	Population	Study Design	Country	Exposure	N	Sex	Age	Outcome(s)	Timing	Main Findings
Burnett PJM, 2022 [17].	Human	RCT	U.K.	Blackcurrant (New Zealand)	13	Healthy male	21 ± 2 years	The running-based anaerobic sprint test The Illinois agility test Seated medicine ball (3 kg) throw Handgrip strength	7 days	Intake of New Zealand blackcurrant extract in rugby union players seems to improve tasks that require maximal speed and agility but not muscle strength. New Zealand blackcurrant extract may be able to enhance exercise performance in team sports that require repeated movements with high intensity and horizontal change of body position without affecting muscle strength.
Yang C, 2022 [18]	Animal	Case–control	China	Wolfberry, figs, white lentils, raspberries, and maca (1.00 g/kg) every day	30	Male mice	NR	Fatigue	30 days	The supplementation of wolfberry, figs, white lentils, raspberries, and maca could improve exercise capacity and relieve fatigue probably by normalizing energy metabolism and attenuating oxidation.
Choi GJ, 2022 [19]	Animal	RCT	South Korea	*Rubus occidentalis*	NR	Adult male Sprague–Dawley rats	NR	Acid-induced hyperalgesia		Immature *R. occidentalis* showed antinociceptive activity against acid-induced chronic muscle pain. Its action may be mediated by the α2-adrenergic, nicotinic cholinergic, and opioid receptors, and immature *Rubus occidentalis* (iROE) displayed a superior antinociceptive tendency to mature ROE.
Hunt JEA, 2021 [20]	Human	RCT	U.K.	Blackcurrant (New Zealand)	27	Healthy and non-resistance-trained males and females	24 ± 2 years	Muscle soreness (using a visual analogue scale) Maximal voluntary contraction (MVC) Range of motion (ROM)	8 days before and 4 days after 60 severe concentric and eccentric contractions of the biceps brachii muscle on an isokinetic dynamometer.	Consumption of New Zealand blackcurrant extract before and after eccentric exercise reduces muscle injury and promotes functional recovery. These findings have practical implications for recreationally active and potentially athletic people, who may benefit from faster recovery after EIMD.
Willems MET, 2020 [21]	Human	RCT	U.K.	Blackcurrant (New Zealand)	12	Male human	24 ± 5 years	Voluntary and twitch force of the quadriceps femoris muscles during repeated isometric contraction-induced fatigue Twitch force during recovery Muscle fiber-specific effects	7 days	New Zealand blackcurrant extract affects force during repeated maximal isometric contractions.
Shukitt-Hale B, 2017 [13]	Animal	Case–control	USA	Red raspberry (*Rubus idaeus*, Meeker variety)	135	Male rats	17 months	Baseline motor performance	8 weeks	Poor performers fed 1% or 2% raspberry showed higher post-test composite scores, whereas 2% raspberry reduced post-test composite scores in good performers as compared to control-fed rats. On plank walking, 1% and 2% raspberry appeared to sustain strong performance while improving poor performance, while 2% raspberry enhanced poor performers’ post-test grip strength. Rats with lower post-diet composite scores had increased blood IL-1β levels.
Costello R, 2020 [22]	Human	RCT	U.K.	Blackcurrant (New Zealand)	20	Male and female recreational runners	30 ± 6 years	Countermovement jumps performance variables Perceived muscle soreness and fatigue	7 days prior to and 2 days following a half-marathon	The Countermovement jump performance characteristics were lowered immediately after the half-marathon, with all reverting to pre-half-marathon levels within 48 h except for concentric and eccentric peak force and eccentric length, with no difference in response between groups. Perceived muscle soreness and fatigue increased immediately after the half-marathon and reverted to pre-marathon levels within 48 h, with no difference between groups.
Fryer S, 2020 [23]	Human	RCT	U.K.	Blackcurrant (New Zealand)	12	Male intermediate rock climbers.	26 ± 5 years	Muscle oxygenation during and following contractions Forearm endurance performance	7 days	There was no difference in time to exhaustion between New Zealand blackcurrant and placebo. During recovery, there was no effect on brachial artery blood flow. However, TTHR was faster with New Zealand blackcurrant following exhaustive contractions.
Brandenburg JP, 2019 [24]	Human	RCT	USA	Blueberry	14	Male and female runners	31.3 ± 10.3 years	Vertical jump The reactive force index	4 days	No significant differences were observed for time to complete the race (TT), heart rate, ratings of perceived exertion, or any of the salivary markers. An interaction effect (*p* = 0.027) was observed for blood lactate, with lower post-TT concentrations in 4DAY than PLA and 2DAY. Post-TT decreases in vertical jump height were not different, whereas the decline in reactive strength index was less following 4DAY (−6.1% ± 13.5%) than the other conditions (PLA: −12.6% ± 10.1%; 2DAY: −11.6% ± 11.5%; *p* = 0.038).
Cook MD, 2017 [25]	Human	RCT	U.K.	Blackcurrant	13	Healthy male	25 ± 4 years	Muscle oxygen saturation Muscle activity Femoral artery diameter of the quadriceps	7 days (14-day washout)	Femoral artery diameter was increased with intake of New Zealand blackcurrant extract during a submaximal (i.e., 30% iMVC) 120 s sustained isometric contraction of the quadriceps muscles. The enlarged diameter of the femoral artery was accompanied by alterations in cardiovascular responses with a decrease in systolic and diastolic blood pressure, mean arterial blood pressure, and total peripheral resistance, with a concomitant increase in cardiac output and stroke volume.
Braakhuis AJ, 2014 [26]	Human	RCT, cross-over	New Zealand	Blackcurrant	23	Female runners	31 ± 8 years	Training progression Incremental running test 5 km time-trial performance	3 blocks of high-intensity training for 3 weeks and 3 days, separated by a washout (~3.7 weeks)	Effects of the two treatments relative to placebo on mean performance in the incremental test and time trial were unclear, but runners faster by 1 SD of peak speed demonstrated a possible improvement on peak running speed with BC juice.
Del Bo’ C, 2012 [27]	Animal	Case–control	Italy	Blueberry	40	Male rats	NR	Vascular smooth muscle contractility and sensitivity	4 or 7 weeks	Wild blueberries incorporated into the diet at 8% *w*/*w* positively affect vascular smooth muscle contractility and sensitivity, but these effects are evident only after 7 weeks of WB consumption.
Norton C, 2005 [28]	Animal	Case–control	USA	Blueberry	30	Male rats	NR	Smooth muscle contractile machinery	13 weeks	Wild blueberries incorporated into the diet affect the vascular smooth muscle contractile machinery by suppressing the α1-adrenergic receptor agonist-mediated contraction while having no effect on membrane sensitivity of the endothelial or vascular smooth muscle cell layer.
Matsumoto H, 2005 [29]	Human	RCT	Japan	Blackcurrant	11	Healthy male	NR	Peripheral muscle circulation	2 weeks	Intake of blackcurrant may improve shoulder stiffness caused by typing work by increasing peripheral blood flow and reducing muscle fatigue.
Yun CE, 2021 [30]	Animal	Case–control	South Korea	Black chokeberry (or aronia)	NR	Wild-type C57BL/6 male mice	8 months	Muscle function and metabolism	8 weeks	The treatment with aronia increases muscle mass and strength in mice without cardiac hypertrophy.
Moghaddam MH, 2021 [31]	Animal	Case–control	Iran	Elderberry	36	Male rats	NR	Motor failure and muscle incoordination	2 months	An elderberry diet significantly recovered motor failure and muscle incoordination in 3-NP-injected rats compared to the control group.

## Data Availability

The original contributions presented in the study are included in the article, further inquiries can be directed to the corresponding author.

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
