# Peer review of "Berry Fruits and Their Improving Potential on Skeletal Muscle Health and Performance: A Systematic Review of the Evidence in Animal and in Human Studies"

_foods, 2024, doi:10.3390/foods13142210_

Round 1
Reviewer 1 Report
Comments and Suggestions for Authors
The authors tried to review on the skeletal muscle improving potential effects of berry fruits. However, the manuscript (MS) has not been ready for publication yet in the present form for the following reasons.
1. For the structure of the MS, it is not reasonable to describe the process of writing the article or obtaining the data in a lot of context space of the RESULTS section. Those in Lines 128-157 should be moved to the section Materials and Methods.
2. The authors simply listed the improvement effects of various berry fruits on skeletal muscle in the RESULTS section. The healthy benefits of berry fruits come from their bioactive compounds. It is important to describe the levels of various bioactive compounds in different berry fruits.
3. It is very important to discuss the mechanism underlying the skeletal muscle improvement of berry fruits in a review paper. This is very poor in the MS.
4. The repeated descriptions in Table 2 and context should be avoided.
Reviewer 2 Report
Comments and Suggestions for Authors
The work needs to be technically refined, for example:
- Word breaks in tables should be avoided where possible.
- In several places in the text the authors have written "aaronia" instead of "aronia" - please correct this.
Page 8 of 19 are empty - please delete.
Also, I would like to ask the authors to check if the other numbers animal models in the text are correctly indicated. For example:
- The abstract states: “Screening of 152 retrieved articles resulted in a final selection of 16 reports investigating the effect of exposure to a berry-based diet and skeletal muscle health outcomes………. Among the selected studies, 9 involved humans and 8 animal models (rats and mice)”. But in Table 2 it is stated that there are 16 models, of which 9 are human and 7 are animal. Therefore, my question is, were there 7 or 8 animal models in the study?
Also:
- In the results – lines 153 to 154 it says: “The included studies were equally distributed between humans (9 out of 17) and animals (8 out of 17).” – why “9 out of 17” and “8 out of 17”? - Why 8 animal models and why 17 if 16 models were involved in the study?
Sentence: "Limitations include the lack of comparison between studies on the same berry and similar samples/models, the small number of studies analyzed, and their heterogeneity in quality and inconsistency of results." in lines 316 to 318 is repeated in lines 321 to 323 – please delete one.
Lines 323 to 330: “In conclusion, considering the positive results on performance or muscle health, the studies suggest that a berry-enriched diet could lead to improvements, but we cannot yet draw specific hypotheses, as the research needs further studies on berries/compounds and specific samples. To sum up, considering the positive results on performance or muscle health, the studies suggest that a berry-enriched diet could lead to improvements, but we cannot yet draw specific hypotheses, as the research needs further studies on berries/compounds and specific samples.” - There are also repetitions of sentences - please correct that.
In chapter “Conclusions” – delete last sentence “This section is not mandatory but can be added to the manuscript if the discussion is unusually long or complex.”
In addition, the search is comprehensive, covers six different databases and follows the guidelines of the PRISMA statement.
The final selection of 16 of the 152 studies identified indicates a rigorous evaluation process. However, it should be noted that only 9 studies were conducted on humans, while other studies were conducted on animal models. The differences between human and animal models may pose a challenge for the generalization of the results.
The results are presented in detail. The main groups of each research group (blackcurrant, blueberry, raspberry, aronia, elderberry and mixed compound) are listed, making it easier to understand the specific effects of different types of fruit on muscle health.
In general, the systematic review provides a comprehensive insight into the current state of research on this topic. The results presented suggest positive effects on muscle performance and health, but it is emphasized that further research is needed.
As far as I can see, the manuscript is solid and a lot of work has been involved in the study.
Round 2
Reviewer 1 Report
Comments and Suggestions for Authors
The concerned issues have been addressed.